∂ | **Open Peer Review** | Microbial Genetics | Research Article

# Genomic trends and emerging antimicrobial resistance in *Neisseria gonorrhoeae* over two decades in Kenya

Supriya D. Mehta,[1,2] Robyn Thorington,[3] Walter Agingu,[4] Fredrick Otieno,[4] Anastasia Unitt,[5] Laura K. Rusie,[2] Adriana Le Van,[6,7] Irene Martin,[3] Ann E. Jerse,[7] Odile Harrison[5]

**ABSTRACT**  We investigated the trends and antimicrobial resistance (AMR) of *Neisseria gonorrhoeae* (NG) in Kenya with whole-genome sequencing (WGS) of isolates collected in 2002–2009 ($n = 108$) and 2020–2022 ($n = 110$). Phenotypic AMR was confirmed by agar dilution. Predicted minimum inhibitory concentrations (MICs), multi-locus sequence typing (MLST), multi-antigen ST (NG-MAST), NG-STAR, and AMR genetic determinants were determined using WGS and detection of molecular markers. The WGS cgMLST typing used LIN codes. Resistance to penicillin, ciprofloxacin, and tetracycline was common throughout. In 2020–2022, azithromycin resistance ($n = 2$) and cephalosporin alert values ($n = 5$) were observed. Phylogenetic clusters were congruent with the LIN code lineage, though other typing schemes (MLST, NG-STAR, and NG-MAST) were not as consistent. There were major shifts over time in the lineages and genetic determinants. Circumcision and HIV status were associated with several AMR, housekeeping, metabolism, and iron acquisition genetic determinants. These findings highlight dynamic NG genomic trends, emerging macrolide resistance, and the value of WGS for surveillance. Behavioral and biological factors may contribute to AMR emergence and warrant further investigation.

**IMPORTANCE** This work highlights the significant value of using whole-genome sequencing to track the evolution and epidemiology of gonorrhea over 20 years: (i) we documented the emergence of azithromycin resistance and cephalosporin reduced susceptibility and relationship to genetics of gonorrhea; (ii) by combining epidemiological and genetic data, we found that circumcision and HIV status were linked to specific genetic features of gonorrhea, including those tied to antibiotic resistance; and (iii) we used novel and traditional genetic typing methods to expand and refine the understanding of lineage shifts and genetic determinants, enhancing surveillance and intervention efforts. Some isolates had potential decreasing susceptibility for cephalosporins, highlighting the critical importance of ongoing surveillance and the opportunity for novel resistance gene identification. Studying how gonorrhea strains relate to a person's immune system, other bacteria (microbiome), and sexual networks could help us understand how certain strains spread and what the potential factors amplifying antimicrobial resistance are.

**KEYWORDS** *Neisseria gonorrhoeae*, gonorrhea, whole-genome sequencing, antibiotic resistance, surveillance studies, human immunodeficiency virus, circumcision, Kenya, Africa, epidemiology

I n 2020, there were an estimated 82.4 million new infections of *Neisseria gonorrhoeae* (NG) worldwide (1), with the highest incidence in sub-Saharan African (SSA) countries (2). Gonorrhea is a clinical and public health threat due to increased risks of HIV acquisition and transmission, long-term impacts on reproductive health, and maternal

Address correspondence to Supriya D. Mehta, Supriyad@uic.edu.

The authors declare no conflict of interest.

See the funding table on p. 13.

gonorrhea, neonatal conjunctivitis, and low birth weight. While gonorrhea is curable with effective antibiotics, antimicrobial resistance (AMR) has been a persistent and growing challenge since the development of penicillin (3).

Surveillance systems, such as the Gonococcal Isolates Surveillance Project (GISP) (4), the European Gonococcal Antimicrobial Surveillance Program (Euro-GASP) (5), GASP-Canada (6), and WHO-GASP (7), provide important information on the prevalence, trends, and types of antimicrobial resistant *N. gonorrhoeae*. This is critical for decision-making around antibiotic treatment regimens. Resistance to penicillin, ciprofloxacin, and tetracycline is frequent among NG isolates from sub-Saharan Africa, though isolates with azithromycin or cephalosporin resistance have been rare (8, 9). However, as THE systematic review of the NG AMR surveillance systems shows, there is a "striking" lack of continuous and systematic surveillance among WHO African Region countries (10). The surveillance gap is particularly pronounced in relation to molecular surveillance. The recent growth in whole-genome sequencing (WGS) to monitor and predict the NG AMR emergence and track transmission across the globe has further widened the surveillance disparity for African countries, which are chronically underrepresented in genomic databases.

HIV remains a global pandemic, with Africa being disproportionately affected. In Kenya in 2020, HIV prevalence among adults aged 15–49 years was estimated at 4.5% nationally and 17.3% in Kisumu, the third largest city in Kenya and the capital of Nyanza Province in western Kenya (11). Voluntary medical male circumcision (VMMC), provid-ing 60% protective effect against HIV acquisition in men (12), has been scaled to an estimated 75–85% of young adult men in western Kenya (13). A putative mechanism by which VMMC reduces risk of HIV acquisition is via changes in the penile microbiome toward a composition that is less inflammatory (14, 15). Host-microbiome interactions influence the risk of NG acquisition and transmission (16), as commensal bacteria may influence transfer and acquisition of resistance genes (17), highlighting the importance of monitoring trends in HIV and male circumcision status in relation to NG and NG-AMR.

Given the high HIV and STI burden in western Kenya, coupled with the limited characterization of NG transmission, AMR prevalence, and AMR emergence that is potentially influenced by host factors, we conducted WGS to understand the genetic relatedness and changing epidemiology of NG in western Kenya in a data set spanning two decades. Our objectives were to: (i) characterize the AMR genetic determinants over time; (ii) compare the NG strains circulating over two decades using novel and conventional typing methods; and (iii) examine the association of individual-level data associated with AMR genes, strains, and phylogeny to gain insight on transmission patterns.

## MATERIALS AND METHODS

### Study design and setting

Data were combined from two studies conducted in Kisumu, Kenya. There were 108 NG isolates collected in 2002–2009 from the randomized controlled trial (RCT) of VMMC (18). Men were aged 18–24 years at enrollment. In addition to nucleic acid amplification testing for NG and *Chlamydia trachomatis* (CT), participants with urethral discharge had an additional swab taken for the NG culture. These isolates were assessed for antibiotic susceptibility by agar dilution (19). Briefly, MICs were determined using the agar dilution method with a GC medium base (Difco Laboratories, Detroit, MI) containing 1% Kellogg's defined supplement and twofold dilutions of antibiotic. *N. gonorrhoeae* ATCC 49226 and World Health Organization (WHO) strains B, C, and D were used as controls. Archived isolates were stored at the National Microbiology Lab (NML) in Winnipeg, Canada.

The 110 NG isolates collected from 2020 to 2022 were from a surveillance study conducted among men aged 18 and older presenting to an STI clinic with complaint of urethral discharge or dysuria. A urethral swab was taken for NG culture and assessed for antibiotic susceptibility by *E*-test, with confirmation of macrolide and ESB resistance

via agar dilution following the Clinical and Laboratory Standards Institute (CLSI) protocol (20). Quality control strains used during all AST protocols were: ATCC 49226, WHO K, WHO L, and an azithromycin-resistant strain obtained from Dr. Olsegun Soge, University of Washington (21). Interpretation cutoffs for resistance and susceptibility were derived from CLSI (22). We compared phenotypic and genotypic AMR for isolates with resistance to azithromycin or minimum inhibitory concentration (MIC) $\geq$ 0.125 for cephalosporins, as this has been associated with resistance emergence (23, 24).

## Whole-genome sequencing methods

### 2002–2009 isolates

All 108 archived isolates from 2002 to 2009 were successfully reanimated. Genomic DNA was extracted using the KingFisher Purification System (Thermo Fisher) and the Omega-Mag Bind Universal DNA Extraction Kit (Omega BIO-TEK). The WGS libraries were prepared using the Nextera XT DNA Library Preparation Kit (Illumina) with sequencing via Illumina NextSeq platform using NextSeq 500/550 v2.5 (300 cycle) chemistry. Eighty-three (76.9%) isolates were successfully sequenced.

### 2020–2022 isolates

A total of 110 NG isolates were sent to the Multidrug Resistant Repository and Surveillance Network (MRSN) for WGS. Methods have been described previously (21), including DNA extraction (DNeasy UltraClean Microbial Kit; Hilden, Germany), library preparation (KAPA Library Quantification Kit; Roche Sequencing, Indianapolis, IN), and sequencing (Illumina MiSeq, NextSeq, or NextSeq 2000 benchtop sequencer (Illumina, Inc., San Diego, CA). Briefly, the *N. gonorrhoeae* taxonomic classification was confirmed *in silico* with Kraken2 (v2.1.2). Further *de novo* draft genome assemblies were constructed with Shovill v1.1.0 (https://github.com/tseemann/shovill), and coverage statistics were assessed with bbmap (v38.96) at minimum thresholds for contig size and coverage set at 200 bp and 49.5×, respectively.

For all isolates (2002–2009 and 2020–2022), the assessment of the sequence quality, assembly, and phylogenetic tree construction was done as previously described (25). The phylogenetic tree was constructed using the Galaxy SNVPhyl pipeline (Galaxy version SNVPhyl v1.0.1b paired-end). Other parameters were as follows: minimum coverage set to 15, minimum mean mapping set to 30, and single nucleotide variant (SNV) abundance ratio set to 0.75, with the removal of highly recombinant regions (>2 SNV per 100 bp). The reference genome used for the phylogenetic tree construction was WHO U.

## Phylogenetic characterization identification of the genetic determinants of AM R

The Cluster Picker program was used to confirm clades using a main support threshold of 0.9, genetic distance threshold of 4.5%, and a large cluster threshold of 10 (26). Predicted MICs, multi-locus sequence types (MLST), multi-antigen sequence typing for *N. gonorrhoeae* (NG-MAST), and NG-STAR types were determined using the R Shiny-Based Application Strep/STI WGS Analysis and Detection of Molecular Markers (WADE) (https://github.com/phac-nml/wade). Genetic determinants are based on those described in the literature (27). The genetic determinants included in WADE are based on the weighted contributions of each element to the overall antimicrobial MICs as determined by multivariate regression (28). The genetic determinants included in WADE are *bla*, *ermB*, *ermC*, *gryA*, *mtrR promoted-region*, *mtrR*, *parC*, *penA*, *ponA*, *porB*, *rpsJ*, *16S rRNA*, *23S rRNA*, and the *tetM* plasmid (Table S1).

## Genomic analyses using PubMLST

Genome assemblies were also uploaded to PubMLST.org (https://pubmlst.org/neisseria), a bacterial sequence database that facilitates the annotation and curation of WGS data

(29). PubMLST is powered by the genomic software, BIGSdb. BIGSdb scans deposited WGS against defined loci, identifying alleles with ≥98% sequence identity and automatically updating isolate records with specific allele numbers, marking regions on assembled contiguous sequences (contigs) for any of the defined loci. Loci are allocated a value-free nomenclature using the prefix "NEIS," followed by four digits, and NG genomes deposited in PubMLST are automatically annotated with any of the defined loci, including those in the typing schemes, that is, MLST, NG MAST, NG STAR, and cgMLST, as well as genes associated with, for example, metabolism and iron acquisition.

## Life identification number (LIN) code analysis

The *N. gonorrhoeae* core genome multi-locus typing scheme version 2 (Ng cgMLST v2) consisting of 1,430 core genes has been implemented in PubMLST and applied in the definition of a LIN code for each isolate: a numeric barcode conveying lineage information in the form of hierarchical clustering at sequential thresholds of allelic mismatch (30). The left-most numbers of the barcode represent the highest thresholds of allelic mismatch and correspond with super-lineage, lineage, and sublineage designations. The rightmost numbers are indicative of highly related isolates, which differ by only a small number of alleles across the 1,430 core genes. All isolates uploaded to PubMLST that meet a minimum of 1,405 of the 1,430 core genes in cgMLST v2.0 are automatically assigned a LIN code. For comparison to full phylogenetic analyses, we generated a maximum likelihood tree based only on the 1,430 core genes. LIN codes were used alongside typing methods, such as MLST, to examine the gonococcal lineages sampled in this analysis. These were compared across years and in association with participant metadata. We also compared the LIN codes of Kisumu isolates to other countries in East Africa defined as members of the East African Community and other sub-Saharan African countries.

## Participant-level data and epidemiologic analyses

Data on age, urban or rural residence, circumcision status, HIV status, and number of sex partners in the past 30 days were collected in both studies. As summarized in Table S1, these were compared by genes known to be associated with AMR, housekeeping genes, select genes associated with iron acquisition (31–33), or with metabolism (31, 34, 35). We targeted iron acquisition markers due to their relevance in vaccine development (16) and metabolic markers that were previously demonstrated to be associated with colonization and virulence (31, 36). In addition, the distributions of MLST, NG MAST, and NG STAR sequence types, and LIN code lineages were compared with epidemiologic factors. Comparisons were made within each time period to separate covariate associations from secular changes in strains and lineages. To avoid sparsity, analyses were restricted to alleles with at least 10 observations (i.e., at least ~5% of the overall sample), with inference only when the comparison sample size remained at least 50% of the overall total (i.e., at least 96 observations). Thus, 41 genes were evaluated. We compared the distribution of genetic markers with the five covariates, stratified by the t–ime period (2002–2009 and 2020–2022). We report the *P*-value from the chi-squared distribution or Fisher's exact test if cells had *n* < 5. Statistical analysis was performed in Stata 18/SE.

## Ethical review

The collection of data, specimens, and NG testing for samples collected from 2002 to 2009 as part of the VMMC RCT was approved by several ethical review committees, as previously detailed (18), and WGS of archived gonococcal isolates and de-identified data were determined to be a non-human subject research. The study involving samples collected from 2020 to 2022 was approved by Maseno University Ethics Review Committee, Kenya, and the University of Illinois Chicago, USA.

**TABLE 1** Distribution of the participant characteristics by the study time period

| Characteristic | 2002–2009 N = 83 | 2020–2022 N = 110 | P-value[a] |
| --- | --- | --- | --- |
| | N (%) | N (%) | |
| Age group (years) | | | <0.001 |
| 18–24 | 83 (100.0) | 48 (43.6) | |
| 25–29 | 0 (0.0) | 34 (30.9) | |
| 30–39 | 0 (0.0) | 22 (20.0) | |
| ≥40 | 0 (0.0) | 5 (4.6) | |
| Missing | 0 (0.0) | 1 (0.9) | |
| Residence | | | 0.06 |
| Rural | 13 (15.7) | 7 (6.4) | |
| Urban | 68 (81.9) | 98 (89.1) | |
| Missing | 2 (2.4) | 5 (4.6) | |
| Circumcision status | | | <0.001 |
| Uncircumcised | 45 (54.2) | 27 (24.6) | |
| Circumcised | 37 (44.6) | 81 (73.6) | |
| Missing | 1 (1.2) | 2 (1.8) | |
| HIV status | | | 0.30 |
| Negative | 73 (88.0) | 99 (90.0) | |
| Positive | 9 (10.8) | 7 (6.4) | |
| Missing | 1 (1.2) | 4 (3.6) | |
| Two or more sex partners in past 30 days | | | <0.01 |
| No | 61 (73.5) | 100 (90.9) | |
| Yes | 19 (22.9) | 8 (7.3) | |
| Missing | 3 (3.6) | 2 (1.8) | |

[a]$\chi^2$ P-value or Fisher's exact if any cell size with $n < 5$.

## RESULTS

One hundred and ninety-three (193) isolates with WGS were analyzed, that is, 83 from 2002 to 2009 and 110 from 2020 to 2022. Age differed between the two study periods as expected because the VMMC trial participation was restricted to men aged 18–24 (Table 1). Isolates from 2020 to 2022 were more likely to come from circumcised men (74% vs 45%, $P < 0.001$). VMMC has been scaled up in Kenya, leading to increased population prevalence of medical male circumcision (14), and in the 2002–2009 period, participants were randomized to VMMC; thus, circumcision was more evenly distributed. Isolates from 2020 to 2022 were from men who were less likely to report multiple sex partners in the prior 30 days (7% vs 23%, $P < 0.01$). HIV prevalence declined over time (10.8% in 2002–2009 vs. 6.4% in 2020–2022), though the difference was not statistically significant.

### Comparison of phenotypic and molecular AMR profiles

Phenotypic antibiotic susceptibility varied over time, with the frequency of isolates exhibiting sensitive (S), intermediate resistance (IR), and resistance (R) summarized in Table 2. As previously reported, resistance to penicillin, ciprofloxacin, and tetracycline was common in both periods (19, 20). While a right-shift in MICs for cefixime, ceftriaxone, and azithromycin was observed in 2002–2009 indicative of reduced susceptibility (19), alert values for cefixime or ceftriaxone (MIC = 0.125, $n = 5$) and resistance for azithromycin (MIC > 2, $n = 2$) were detected only in the 2020–2022 period (Table 3) (20).

Table 3 summarizes the results of agar dilution and WGS (genotype) for isolates with phenotypic resistance to azithromycin and elevated MICs for ceftriaxone or cefixime (37). Unsurprisingly, the two isolates with azithromycin resistance (agar dilution MIC ≥ 256) had the A2059 mutation in all four 23S rRNA alleles and high predicted MICs; this was not observed in any other isolate, highlighting the specificity of this finding. For the five isolates with elevated MICs for cephalosporin, the predicted MICs were all ≤0.031. The genetic markers A516G and A501V in *penA* that are typically associated with

**TABLE 2** Phenotypic antibiotic susceptibility of gonococcal isolates collected from symptomatic men in Kisumu, Kenya in 2002–2009 and 2020–2022 following the CLSI classification

| Antimicrobial agent | 2002–2009, $N = 82^{a,b}$ | | | 2020–2022, $N = 110$ | | |
| | S | IR | R | S | IR | R |
| | n (%) | n (%) | n (%) | n (%) | n (%) | n (%) |
|---|---|---|---|---|---|---|
| Tetracycline | | 1 (1.2) | 81 (98.8) | 3 (2.7) | 5 (4.5) | 102 (92.7) |
| Ceftriaxone | 82 (100) | | | 110 (100) | | |
| Gentamicin[c] | | | | 84 (76.4) | 26 (23.6) | |
| Penicillin | 28 (34.1) | | 54 (65.9) | 10 (9.1) | 20 (18.2) | 80 (72.7) |
| Ciprofloxacin | 69 (84.1) | | 13 (15.9) | 5 (4.5) | 15 (13.6) | 90 (81.8) |
| Azithromycin | 82 (100) | | | 108 (98.2) | | 2 (1.8) |
| Cefixime | 82 (100) | | | 110 (100) | | |
| Spectinomycin | 82 (100) | | | 110 (100) | | |

[a]Antimicrobial susceptibility was determined by agar dilution for isolates collected in 2002–2009 and by E-test with agar dilution to confirm elevated MICs for isolates collected in 2020–2022.
[b]MIC from 2002 to 2009 for one isolate could not be located.
[c]Gentamicin susceptibility was not assessed in isolates collected 2002–2009.

increased MIC to cephalosporins were observed. We repeated agar dilution testing of these samples and obtained the same MICs.

## Phylogenetic strain typing

Of the 193 isolates in this analysis, 152 were included in phylogenetic analyses (Fig. 1) with lack of depth in genomic coverage excluding 41 NG WGS samples. Phylogenetic analyses separated isolates into two large clusters, with further sub-clusters observed within each of these. The LIN code superlineages and lineages were congruent with phylogenetic clusters, in particular two large clades ($n = ≥10$) identified in box 1 (blue) and box 2 (green) consisting of NG from lineages 0_0_36 ($n = 16$) and 0_0_93 ($n = 11$). The other typing schemes comprising MLST, NG-STAR, and NG-MAST were not as congruent with the LIN code lineages most likely as a result of the horizontal genetic transfer in the small numbers of genes used in these typing schemes. The tree separated isolates temporally, with most (15/16, 94%) clade 1 isolates dating from 2002 to 2009, with one isolate from 2021 for which a LIN code was not assigned. All clade two isolates were dated from 2020 to 2022. The lower section of the tree was more genetically diverse, with no clades identified containing more than four isolates. While there were some LIN code lineages in this portion of the tree that had three or more samples associated with them, the average number of samples associated with a LIN code lineage in the bottom half of the tree is 6.3 compared to 11.75 in the remainder of the tree. The phylogenetic tree using the cgMLST scheme showed clustering and alignment to LIN code lineage and traditional typing schemes (Fig. S1).

## LIN code analysis

LIN codes were assigned to 183/193 isolates, comprising 34 different LIN code lineages (i.e., 34 clusters of isolates differing in no more than 300/1430 core genes). Of these 183 isolates, the most common lineages were 0_0_36 (20.2%) and 0_0_33 (14.8%), followed by 0_0_47 (7.1%) and 0_0_93 (6.0%) (Fig. 2). Therefore, 48.1% of the isolates belonged to only four lineages, all within superlineage 0_0. The representation and proportion of lineages varied over time, with 2002–2009 harboring the majority of 0_0_36 isolates (28/37, 81.5%), while all 0_0_33 lineage isolates were observed in 2020–2022. Similarly, the majority of lineage 0_0_93 isolates (10/11, 90.9%) were observed in 2020–2022. No isolates were detected belonging to the globally dominant lineages 0_2_1 and 0_2_0.

## Comparison to other African isolates

At the time of analysis (November 2024), PubMLST contained 1,173 African isolates from outside this study, and 820 (69.9%) were assigned a LIN code. Isolates lacking a LIN code

**TABLE 3** Comparison of genotypic and phenotypic AMR for isolates collected in 2020–2022 exhibiting resistance to azithromycin or elevated minimum inhibitory concentration for ceftriaxone or cefixime[c,d]

| Isolate | AZI AD MIC (µ/mL) | AZI predicted MIC (µ/mL) | mtr[b] | 23S_rRNA alleles[a] | CRO AD MIC (µ/mL) | CRO predicted MIC (µ/mL) | CFX AD MIC (µ/mL) | CFX Predicted MIC (µ/mL) | NEIS173 (penA) | NEIS2020 (porB) G120/A121 | NEIS0414 (ponA) | MLST/ LINcode |
|---|---|---|---|---|---|---|---|---|---|---|---|---|
| KS002 | 0.047 | ≤0.12 | A39T | 0 | 0.0625 | 0.008 | 0.125 | ≤0.008 | Non-mosaic 14.001 A517G/ | porB1b WT/WT | WT | ST-1588 0_0_89_0_1 |
| KS017 | 0.094 | ≤0.12 | A39T | 0 | 0.125 | 0.008 | 0.125 | ≤0.008 | Non-mosaic 9.001 A517G/ | porB1b WT/WT | L421P | ST-1902 0_0_93_0_0 |
| KS037 | 0.25 | ≤0.125 | −35Adel | 0 | 0.25 | 0.031 | 0.125 | 0.016 | Non-mosaic 11.001 A502V/ A517G | porB1a NA/NA | L421P | ST-8134 0_0_47_5_15 |
| KS041 | 0.125 | ≤0.12 | −35Adel | 0 | 0.125 | 0.031 | 0.0625 | 0.016 | Non-mosaic 11.001 A502V/ A517G | porB1a NA/NA | L421P | ST-8134 0_0_47_5_15 |
| KS066 | 0.25 | ≤0.12 | −35Adel | 0 | 0.125 | 0.031 | 0.0625 | 0.031 | Non-mosaic 11.001 A502V/ A517G | porB1a NA/NA | L421P | ST-8134 0_0_47_5_15 |
| KS087 | ≥256 | ≥256 | A39T | A2059x4 | <0.016 | 0.016 | <0.016 | 0.016 | Non-mosaic 2.002 A517G/ | porB1b G120K/A121N | WT | ST-11994 1_1_54_0_2 |
| KS100 | ≥256 | ≥128 | WT | A2059x4 | <0.016 | 0.008 | <0.016 | ≤0.008 | Non-mosaic 2.002 A517G | porB1b WT/WT | WT | ST-11994 1_1_54_0_1 |

[a]Other resistance markers for macrolides (23S rRNA 2611T, and *mtrD*) were not present.
[b]*mtr*: includes promoter and *mtr* gene; all were WT for G45.
[c]NA, algorithm unable to determine.
[d]AZI, azithromycin; CRO, ceftriaxone; CFX, cefixime; AD, agar dilution; MIC, minimum inhibitory concentration.

tend to represent poor quality assemblies (defined as not having a minimum of 1,405/1,430 core genes annotated), with large numbers of short contigs, which interrupt core loci. Among the 820 other African isolates with a LIN code, 617 (75.2%) were East African, stemming from Kenya ($n = 191$), Uganda ($n = 424$), and Tanzania ($n = 2$). There were no Kenyan isolates from outside of our study between 2002 and 2009 that had a LIN code assigned; therefore, we compared our 2002–2009 isolates to 76 Kenyan isolates collected 2010–2012 (Fig. 2A). There were only three Kenyan isolates collected outside of our study between 2020 and 2022 with a LIN code assigned; therefore, we compared our 2020–2022 isolates to those collected in 2016–2020 (Fig. 2, Panel A). Of the 424 isolates from Uganda, 423 were collected in 2018, so we compared these to our 2020–2022 isolates (Fig. 2A and B). As shown in Fig. 2, the distribution of lineages was strikingly different for isolates collected in other Kenyan sites in 2010–2012 as compared to our Kisumu 2002–2009 isolates. Conversely, the distribution of lineages for isolates collected in Kisumu in 2020–2022, other Kenyan sites in 2018–2020, and Uganda in 2018 was largely similar, with the majority being 0_0_33 or 0_0_36.

Given the small sample sizes for each lineage when stratified by region and time period, we directly compared only the superlineage distributions for SSA vs. Kisumu, where superlineage is indicative of isolate clustering at a threshold of >600 locus differences. In 2002–2009 (Fig. 2B), other SSA gonococcal isolates (all from Guinea Bissau) had a high frequency of superlineage 1_1 (62.5%) compared to Kisumu's higher frequency of 0_0 (53.4%). The 2020–2022 distribution of superlineages showed similarities

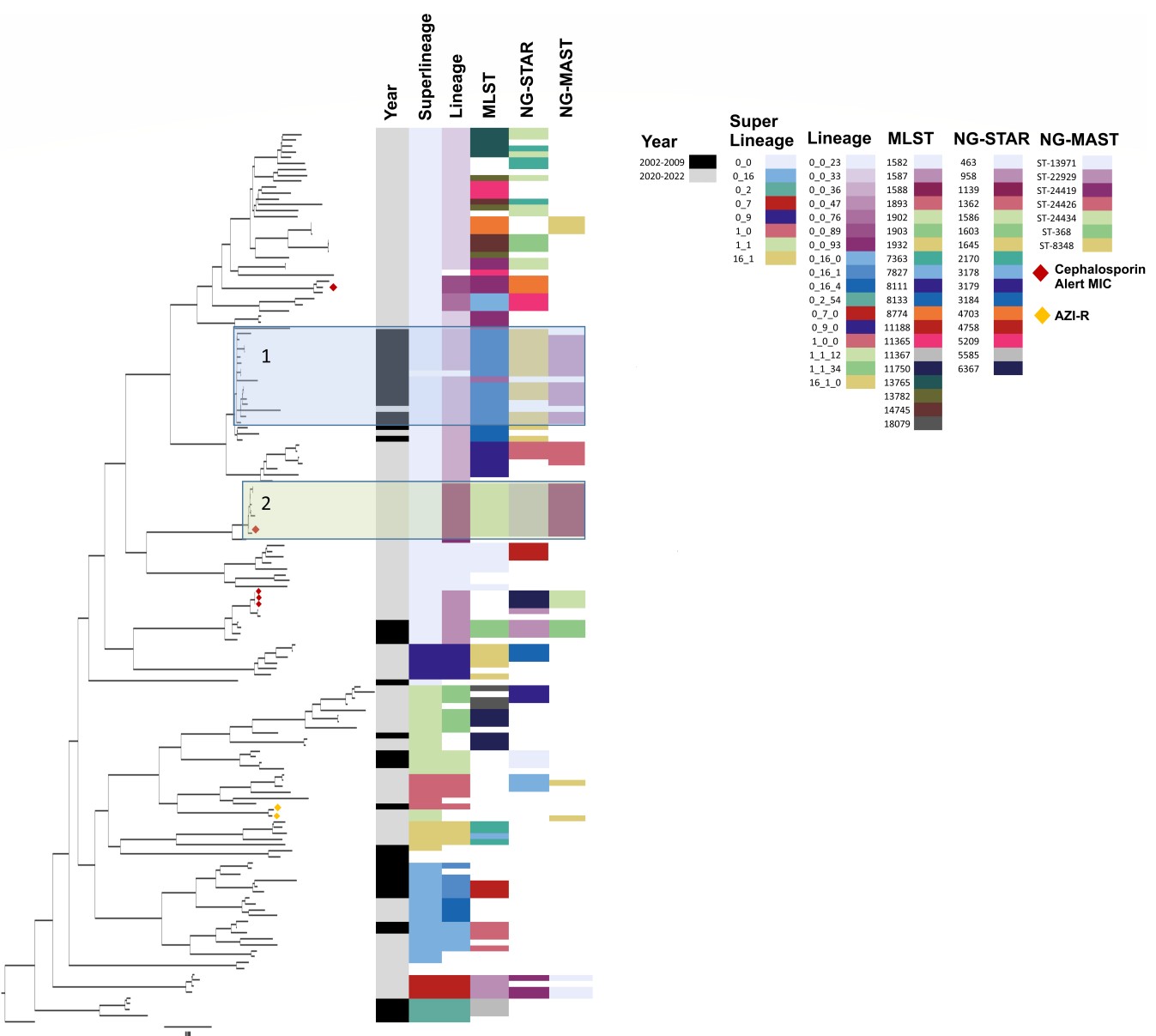

**FIG 1** Phylogenetic tree representing 152 gonococcal isolates with demarcation of four clades and alignment to the year of isolation, LIN lineage and sublineage, and standard classifications. Legend: the phylogenetic tree with two clades demarcated (Box 1 [blue] and Box 2 [green]) and alignment of each isolate to the year of isolation, LIN codes (lineage and sublineage), and traditional classification schemes. The two isolates exhibiting azithromycin resistance are indicated with yellow dots, and the five isolates exhibiting cephalosporin alert values are indicated with red dots. LIN code lineages with less than three isolates are indicated in white. If three or more samples had the same LIN code or sequence type, then that molecular type was assigned a color in the legend of the figure.

for Kisumu and Uganda 2018, though 0_0 was more common for Kisumu and 0_15 more common for Uganda. The 2020–2022 SSA isolates represented Cameroon (*n* = 7), Kenya (*n* = 3), Madagascar (*n* = 25), and Zambia (*n* = 2). The distributions were quite different, with many isolates from Madagascar, which was the sole source of 1_2 and 0_24 super lineages.

**A) Distribution of Lineage for Gonococcal Isolates from Kisumu, Other Kenya Sites, and Uganda, Stratified by Time**

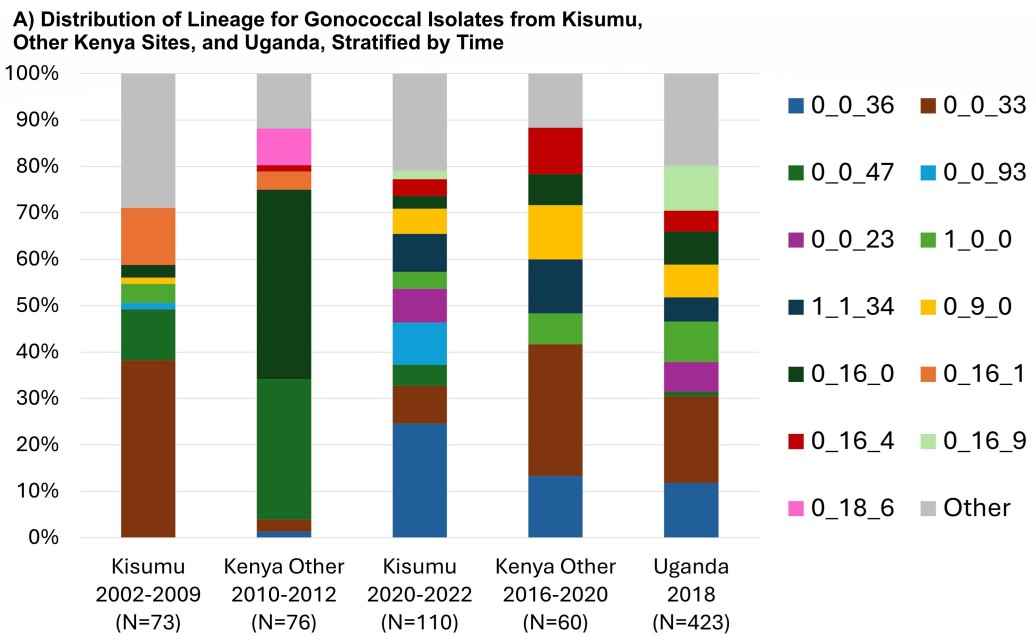

**B) Distribution of Superlineage for Kisumu Isolates and Other Sub-Saharan African Gonococcal Isolates, Stratified by Time**

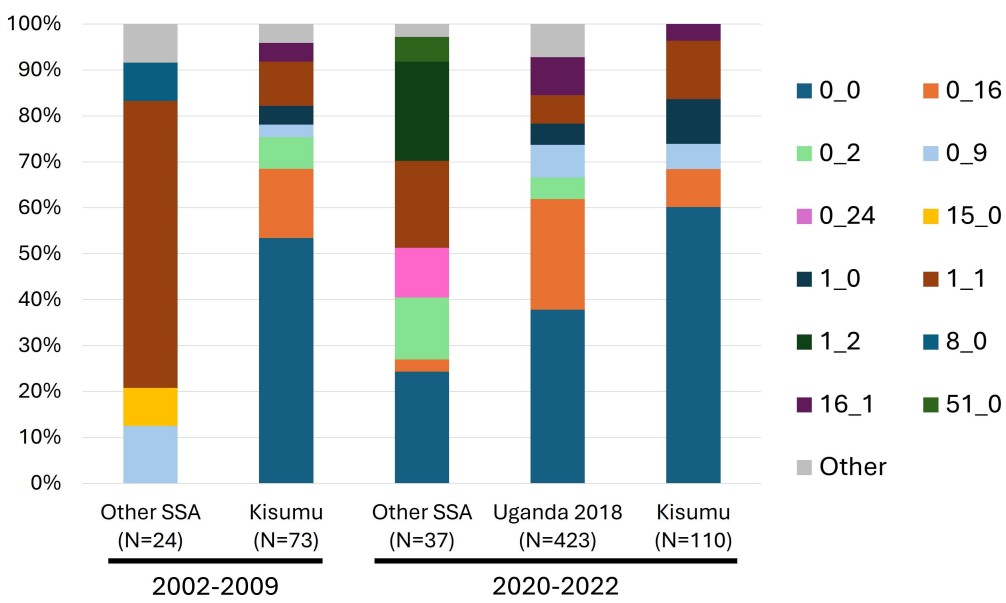

**FIG 2** Distribution of the LIN lineage and superlineage for gonococcal isolates from Kisumu stratified by time and compared to: (A) other Kenyan sites and Uganda, and (B) sub-Saharan Africa.

## Association of molecular markers with circumcision status, HIV status, and age

Figure 3 summarizes the participant level data and allele distributions that differed with a $P < 0.05$ level: circumcision status, HIV status, and age. No significant differences were observed for urban vs. rural residence or number of sex partners. In both time periods, NG from circumcised men ($n = 117$) more frequently possessed the MLST *abcZ* allele 126 ($n = 9$) and the MLST *pdhC* allele 153 ($n = 9$). These belonged to ST-1903 ($n = 5$), ST-8134 ($n = 3$), ST-12241 ($n = 1$), and ST-13943 ($n = 1$). NG from circumcised men more frequently exhibited the −35Adel of the *mtrR* promoter mutations in both time periods, though this difference was significant at the $P < 0.05$ level only in 2002–2009. NG from

circumcised men more frequently exhibited NG STAR *ponA* allele 1 (L421P), *porB* allele 100 (WT), and *porB1a* in 2002-2009, but these differences were not observed in 2020–2022. Numerous NG genes varied by HIV status within the time periods. This included the MLST genetic determinants *abcZ* and *aroE*, the iron acquisition genes *hpuAB* (NEIS1946 and NEIS1947) and *fetA* (NEIS1963), and the genes *ppk* (NEIS0323; encoding a polyphosphate kinase) and *pykA* (NEIS0074; encoding pyruvate kinase in the glycolysis pathway). In 2020–2022, compared to isolates from HIV negative participants, isolates from HIV positive participants more frequently possessed MLST *abcZ* allele 126, MLST *aroE* allele 170, *gyrA* genes harboring the amino acid substitution D95G conferring resistance to ciprofloxacin, allele 7 of NEIS0936 (*sucD*) (part of the citric acid cycle and encoding succinyl-CoA synthetase subunit alpha), and a different allelic profile for *hpuA*. Among 2020–2022 isolates, HIV-infected participants frequently had NG from lineages 0_0_36 and 1_1_34 compared to HIV-uninfected participants. We limited comparisons of age to men aged 18–24 since the 2002–2009 period of the VMMC trial was limited to this age range. Isolates from 18 to 24 year-olds more frequently exhibited *pykA* allele 12 and allele 474 in 2020–2022. Lineage and sub-lineage varied over time in 18–24 year-olds, with a higher frequency of 0_0_93 and 0_0_33 in isolates recovered in 2002–2009 compared to 2020–2022, which had higher frequency of 0_0_47 isolates. Sublineages of isolates collected from 18 to 24 year-olds were strikingly dissimilar between time periods, and the number of comparable alleles was sparse.

## DISCUSSION

We sought to understand the genetic relatedness and the changing epidemiology of *N. gonorrhoeae* in western Kenya in a data set spanning two decades, with an emphasis on antibiotic resistance determinant alleles. WGS of NG collected in Kisumu, Kenya in 2002–2009 and 2020–2022 revealed how the bacterial lineages sampled changed over time, as did the genetic determinants and mechanisms associated with macrolide resistance. These genetic determinants and lineages varied according to host characteristics, most commonly circumcision and HIV status. Five isolates exhibited MICs above alert values for cephalosporins; although WGS did not predict cephalosporin MIC alert using WADE, commonly described genetic factors associated with cephalosporin MIC creep were identified. The LIN code analysis, a novel lineage nomenclature for classifying NG based on the core genome MLST, showed good comparability to phylogenetic analyses while being able to annotate WGS of lower quality.

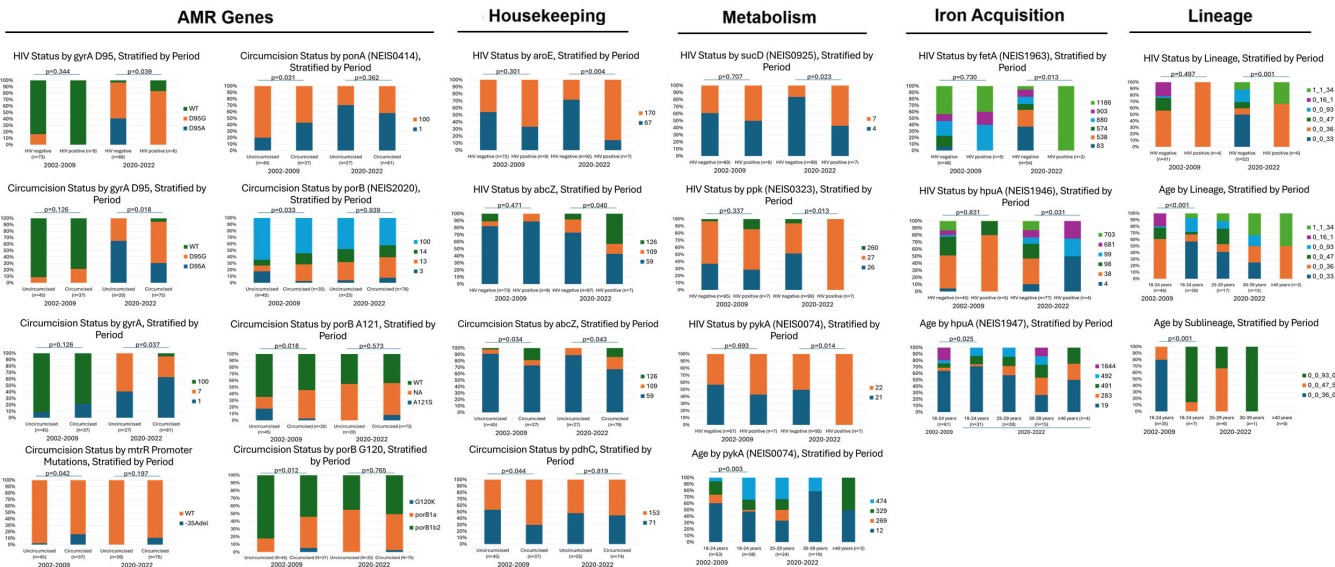

**FIG 3** Distribution of alleles by participant covariates, stratified by the time period.

The LIN code analysis revealed that the strains circulating in Kisumu were largely similar to other Kenyan isolates and those from neighboring Uganda but differed from those of broader sub-Saharan Africa, though a comparison may not be valid due to the limited numbers of time-matched isolates available for comparison. Notably, lineage 0_0_33 (representing 14.8% of our isolates) is widely geographically distributed, being in the top 20 gonococcal lineages of all six continents. Meanwhile, lineage 0_0_36 (representing 20.2% of our isolates) is more common in Africa (9.0% of isolates) and rare in other continents (Europe 3/14,943 [0.02%] and Oceania 1/3,272 [0.03%]). Therefore, the NG sampled from Kisumu belongs to lineages associated both with globally distributed and Africa-specific clades.

However, Kenyan isolates originating from 2010 to 2012 harbored notably different lineages. Meta-data for these isolates in PubMLST are limited but show that 10 were sourced from the female reproductive tract in central Kenya, and the remainder primarily from men who have sex with men in coastal Kenya. These geographically and epidemiologically different source populations may explain our observation and highlight the need for diverse source populations for surveillance, as they likely have varying sexual networks.

LIN code lineages showed a good level of agreement with phylogenetic analyses and traditional classification schemes,such as the seven-loci MLST, for certain clades. It was visible on the phylogeny that some MLST STs were distributed across the tree; this incongruence is likely due to the horizontal gene transfer in the seven housekeeping genes used in MLST. This problem when applying the seven-loci MLST to gonococci has been noted previously (38). In comparison, the LIN code, which applies a much larger number of loci, was more reliably able to replicate the clades observed in the phylogenetic tree. Our supplementary tree based on the cgMLST 1430 core genes included all 183 isolates; thus, thes lack of depth in genome coverage was not such a problem. At the same time, this tree and alignment to LIN code lineages and traditional typing methods were largely similar to the full-genome phylogenetic analysis, underscoring the robustness of the results and highlighting an advantage of this approach.

The two azithromycin-resistant isolates (KS087, ID 114036 and KS100, ID 144049) belonged to ST-11994 and lineage 1_1_54; they were the only two isolates with these classifications in our study and the entire SSA lineage data set. Based on MLST (ST-11994; *abc* 59, *adk* 39, *aroe* 67, *fumc* 157, *gdh* 769, *pdhc* 153, *pgm* 65), PubMLST harbors five other similar isolates originating from Portugal (ID 87791 and ID 163667, both 2013), Spain (ID 42104, 2013), Malawi (ID 161913, 2016), and the United Kingdom (ID 132042, 2022). The Malawian isolate was the only other isolate in the database belonging to lineage 1_1_54, while the isolates from Portugal were both from lineage 1_1_0 (the UK and Spanish isolates did not have LIN codes assigned). In fact, the Malawian isolate and both Kenyan isolates belonged to the same sublineage, 1_1_54_0 (meaning they differ in no more than 125 loci out of the 1,430 core genes), although their LIN barcodes diverged after this point. This trio of isolates may, therefore, be related. However, the Malawian isolate was not azithromycin-resistant (39).

Five isolates exhibited MICs with alert values for cephalosporins but not resistance; even though these did not carry mosaic *penA* alleles, they had mutations A502V and A517G, which have been reported to decrease susceptibility to cephalosporins (40, 41). Genomic study of 118 U.S. GISP isolates collected in 2009–2010 with reduced susceptibility to cefixime revealed a high recombination-to-mutation ratio. While nearly all harbored *penA* mosaic alleles, not all did, and authors surmise there are other factors determining cefixime MIC (42). It is also possible that these reduced susceptibilities are not stable: within phylogenetic clade, some isolates were susceptible, suggesting reversion as a possibility (42). Genomic study of U.S. GISP isolates also did not find perfect concordance between phenotype and genotype for cefixime, though the potential source of discrepancy was not discussed (43). Three of these isolates belonged to the same ST (ST-8134), and further analysis using LIN codes demonstrated that they were highly related. Isolates KS037 (ID 143986) and KS066 (ID 144015) shared a complete

LIN barcode (0_0_47_5_15_0_0_0_0_1), meaning they were identical across their core genome, while isolate KS041 (ID 143990) differed at the last threshold only, meaning it shared all but one core genome allele. This implies these isolates may represent a transmission chain. These LIN codes are not of a lineage usually associated with AMR; resistance to cephalosporins is more commonly associated with isolates belonging to LIN lineage 0_2_0. Our isolates with MIC alert levels for cephalosporin were observed only in the 2020–2022 period, highlighting the critical importance of ongoing surveillance. Additionally, these isolates can be mined for novel resistance genes.

The individual-level data most frequently associated with *N. gonorrhoeae* genetic markers were male circumcision status and HIV status. The main features differentiating medically circumcised and uncircumcised men are penile microbiome composition and sociodemographics. Therefore, we hypothesize the associations we observed with NG genetic markers could represent potential mechanisms related to (i) penile microbiome and immunology and (ii) sexual networks. For adult men, VMMC results in a significant shift in the penile microbiome composition with a reduction in anaerobic bacteria (44, 45), reduced mucosal inflammation (14), and altered immune cell architecture (15). The relationship between genital microbiome and AMR has not been studied, though enhanced AMR gene expression has been demonstrated among hospitalized patients with dysbiotic gut microbiome (46). Penile microbiome composition, through its influence on eliciting a polymorphonuclear leukocyte (PMN) response, may also contribute to NG selection and transmission through killing of more susceptible strains (36, 47). Additionally, it has been demonstrated that vaginal microbiota composition may enhance *N. gonorrhoeae* transmissibility (48), though this has not been investigated for the penile microbiome. Greater abundance and load of anaerobic and inflammatory penile bacteria are associated with HIV seroconversion; thus, differences observed in relation to the HIV status may overlap with differences in penile microbiome composition or mucosal immunology. As the penile microbiome may be modifiable (49, 50), our results provide rationale for investigating the hypothesized relationship between penile microbiome and NG AMR. Alternatively, differences in *N. gonorrhoeae* strains by circumcision and HIV status may be related to sexual network composition, as phylogenetic studies demonstrate NG clustering by HIV status and sexual practices (e.g., men who have sex with men) (51, 52). Additionally, men who remain uncircumcised have different sociodemographic features from circumcised men—being older, with less educational attainment and more economic barriers (53). Given tendencies to homophily among sexual partner selection, men remaining uncircumcised are likely to have different sexual networks than circumcised men (54). There are few studies combining *N. gonorrhoeae* phylogenetic analyses with direct rather than inferred measure of sexual networks, which could advance knowledge of how NG strains circulate within communities, potentially amplifying AMR emergence and spread.

## Limitations

The geographic comparison of the LIN codes was hampered by the limited number and representation of the sub-Saharan African isolates by time point. Notably, although there were 1,395 isolates from SSA, they represent less than 5% of the more than 33,593 *N. gonorrhoeae* isolates publicly available in PubMLST as of March 2025. Valid and reliable inference on the strains and epidemiology of NG in SSA and contextualization to the global epidemiology will be obscured until there is comprehensive and ongoing NG surveillance. Collection of epidemiological data within PubMLST is often unique to the projects, impeding assessment of host characteristics that may help explain differences over time or across locations. In our analysis, the inclusion of expanded meta-data supports novel hypothesis generation regarding the potential biological or sexual network interactions with *Neisseria gonorrhoeae*.

## Conclusions

The whole-genome sequencing of *Neisseria gonorrhoeae* isolates collected in Kisumu, Kenya over a 20-year period shed light on and underscored the lack thereof of the genetic determinants of AMR emergence in African NG. We demonstrate that LIN codes are a useful adjunct to surveillance and for epidemiologic analyses applied here to describe the similarity and differences in NG within Kenya and SSA. Our analyses illuminated novel hypotheses to explore as potential drivers of AMR emergence or transmission, namely, male circumcision and HIV statuses, and highlighted the importance of epidemiologic measures in surveillance data.

## ACKNOWLEDGMENTS

We thank Ezekiel Dibondo and Reuel Sandy for technical work. We also thank the initial investigators from the randomized trial of voluntary medical male circumcision in Kisumu: Kawango Agot (deceased), Robert Bailey, Ian Maclean, Stephen Moses, and Jeckoniah Ndinya-Achola (deceased).

This research was supported by grant AI50440 from the Division of AIDS, National Institute of Allergies and Infectious Disease of the U.S. National Institutes of Health (NIH), by AI50440-S (American Recovery and Reinvestment Act), by grant HCT 44180 from the Canadian Institutes of Health Research (CIHR), and by the Henry M. Jackson Foundation (HU0001-19-2-0068). The funders had no role in the design of the study, the collection, analysis, and interpretation of data, or in writing the manuscript.

## AUTHOR AFFILIATIONS

[1]Division of Epidemiology & Biostatistics, University of Illinois Chicago School of Public Health, Chicago, Illinois, USA
[2]Division of Infectious Disease Medicine, Rush University Medical College, Chicago, Illinois, USA
[3]National Microbiology Laboratory Branch, Public Health Agency of Canada, Winnipeg, Canada
[4]Nyanza Reproductive Health Society, Kisumu, Kenya
[5]Nuffield Department of Population Health, University of Oxford, Oxford, United Kingdom
[6]Henry M. Jackson Foundation for the Advancement of Military Medicine, Bethesda, Maryland, USA
[7]Uniformed Services University, Bethesda, Maryland, USA

## AUTHOR ORCIDs

Supriya D. Mehta ⓘ http://orcid.org/0000-0002-7926-2489
Irene Martin ⓘ http://orcid.org/0000-0002-3941-5583
Odile Harrison ⓘ http://orcid.org/0000-0002-1623-0295

## FUNDING

| Funder | Grant(s) | Author(s) |
| --- | --- | --- |
| National Institutes of Health | AI50440, AI50440-S | Supriya D. Mehta |
| Canadian Institutes of Health Research | HCT 44180 | Supriya D. Mehta |
| Henry M. Jackson Foundation | HU0001-19-2-0068 | Supriya D. Mehta |
| | | Walter Agingu |
| | | Fredrick Otieno |
| | | Adriana Le Van |
| | | Ann E. Jerse |

## AUTHOR CONTRIBUTIONS

Supriya D. Mehta, Conceptualization, Data curation, Formal analysis, Funding acquisition, Investigation, Methodology, Project administration, Visualization, Writing – original draft, Writing – review and editing | Robyn Thorington, Data curation, Formal analysis, Investigation, Methodology, Validation, Visualization, Writing – original draft, Writing – review and editing | Walter Agingu, Data curation, Investigation, Writing – review and editing | Fredrick Otieno, Data curation, Investigation, Project administration, Supervision, Writing – review and editing | Anastasia Unitt, Formal analysis, Methodology, Visualization, Writing – review and editing | Laura K. Rusie, Formal analysis, Visualization, Writing – review and editing | Adriana Le Van, Data curation, Investigation, Methodology, Project administration, Resources, Validation, Writing – review and editing | Irene Martin, Data curation, Investigation, Methodology, Project administration, Resources, Supervision, Writing – review and editing | Ann E. Jerse, Data curation, Investigation, Methodology, Resources, Writing – review and editing | Odile Harrison, Data curation, Formal analysis, Investigation, Methodology, Resources, Software, Supervision, Writing – original draft, Writing – review and editing

## DATA AVAILABILITY

The data used in this analysis (meta-data and whole genome sequencing data) are available in PubMLST isolate ids #143950–143999, 144000–144059, and 149153–149234 (https://pubmlst.org/neisseria). Additionally, 2002–2009 isolates are available at: https://pathogen.watch/collection/8esi6yhv8fax-kisumu-2002-2009 and 2020–2022 isolates are available at: https://pathogen.watch/collection/0bhfemrahe5j-kisumu-2022.

## ADDITIONAL FILES

The following material is available online.

### Supplemental Material

**Fig. S1 (Spectrum01586-25-s0001.docx).** Phylogenetic (maximum likelihood) tree based on the Ng cgMLST v2 scheme demonstrating alignment to the LIN code lineage and conventional typing schemes.
**Table S1 (Spectrum01586-25-s0002.docx).** List of genetic determinants and associated antimicrobial resistance or mechanism that was evaluated in the analysis.

### Open Peer Review

**PEER REVIEW HISTORY (review-history.pdf).** An accounting of the reviewer comments and feedback.

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
