## [Reviewer comments · Microbiology Spectrum]

Microbiology Spectrum

Genomic Trends and Emerging Antimicrobial Resistance in *Neisseria gonorrhoeae* Over Two Decades in Kenya

Supriya Mehta, Robyn Thorington, Walter Agingu, Fredrick Otieno, Anastasia Unitt, Laura Rusie, Adriana Le Van, Irene Martin, Ann Jerse, and Odile Harrison

Corresponding Author(s): Supriya Mehta, University of Illinois Chicago

Review Timeline:

Submission Date:	May 21, 2025
Editorial Decision:	July 4, 2025
Revision Received:	August 5, 2025
Accepted:	September 1, 2025

Editor: Monica Cartelle Gestal

Reviewer(s): The reviewers have opted to remain anonymous.

Transaction Report:

DOI: <https://doi.org/10.1128/spectrum.01586-25>

Re: Spectrum01586-25 (**Genomic Evolution and Emerging Antimicrobial Resistance in *Neisseria gonorrhoeae* Over Two Decades in Kenya**)

Dear Dr. Supriya D Mehta:

Thank you for the privilege of reviewing your work. Below you will find my comments, instructions from the Spectrum editorial office, and the reviewer comments.

This manuscript Spectrum01586-25 by Supriya et al reports the genetic variability and epidemiology associated with *Neisseria* infection in Kenya during a 20 years study period. Both reviewers agree on the importance of this manuscript and a revision is necessary. Result section can be tedious at the times, please try to not only provide the data but tell a story that engages the reader. Maybe breaking result sections into smaller more focus sections would be helpful. Figure 1 is extremely difficult to read, maybe increasing the size font would be helpful to better understand this figure, which is complex due to the great amount of information herein included. This comment is also applicable to figure 3.

Please carefully address the reviewers comments and provide a track changes revised version.

Revision Guidelines

Sincerely,
Monica Cartelle Gestal
Editor
Microbiology Spectrum

Reviewer #1 (Comments for the Author):

This study by Mehta et al. is a well-designed, technically sound, and epidemiologically insightful analysis of AMR in Kisumu, Kenya. Overall, the study provides significant insights into genomic evolution and the AMR emergence of *N. gonorrhoeae* in the sub-Saharan African region using novel technology. However, the study is limited by sample collection in a limited geographical location and the lack of direct biological and behavioral data supporting some hypotheses. These gaps are essential to address in follow-up studies. Nonetheless, the study provides a useful longitudinal genomic analysis of *N. gonorrhoeae* in Kenya, uncovering emerging resistance, linking host factors to AMR evolution, and promoting genomic tools like LIN code for epidemiological surveillance. Below are several strengths and limitations for future consideration:

- The high-resolution approach allows identification of genetic determinants of AMR and lineage tracking.
- The integration of cgMLST and LIN codes improves the resolution and comparability of isolates beyond traditional methods.
- Two-decade comparison (2002-2009 vs 2020-2022) offers a rare longitudinal insight into AMR evolution.
- Comparison with other Kenyan sites and sub-Saharan isolates adds a valuable regional comparison.
- Analysis of HIV status, circumcision status, and age links patients' data to microbial evolution, a novel and valuable epidemiologic dimension.
- Documentation of azithromycin resistance and cephalosporin alert-level MICs is a significant development.
- The authors mention sexual network composition as a possible determinant for the evolution of *N. gonorrhoeae* strains, but social-behavioral data (e.g., partner characteristics, MSM status, condom use) were not collected or analyzed, limiting causal analysis.
- The study proposes that penile microbiome shifts due to VMMC or host inflammation may influence AMR evolution, but this remains highly speculative, as there is no microbiome data, no immunological data, and no causal modeling.

Reviewer #2 (Comments for the Author):

Manuscript by Mehta et al., describes genetic determinants of *N. gonorrhoeae* (NG) antimicrobial resistance in Kenya over two decades. The authors examined the changes in circulating NG strains in central regions of Kenya and assessed the possible links between levels of AMR levels, NG type and HIV infection in 193 NG isolates collected between 2002-2009 and then 2020-2022.

While the topic is important and the study adds regional data on NG AMR trends, limitations in data analysis makes it hard to interpret the findings. The authors do not establish specific links between observed AMR phenotypes and the emergence of known resistance-conferring mutations in the tested isolates. Additionally, the authors did not provide broader insight into other genetic adaptation across the timepoints. Only a relatively small group of related isolates is analyzed in depth. For example, in Fig. 1, there are many isolates between two cohorts that share similar ancestral isolate, but they are not discussed, and the genetic/phenotypic changes are not discussed.

In general, the main findings are not clear and the manuscript provides descriptive survey of NG strain types and AMR profiles than a thorough genomic or evolutionary analysis. A deeper and more robust analysis of genomic data, other than MLST, phylogeny, and mutations linked to AMR could provide valuable information on pathogenesis, AMR and local epidemiological trends.

The method section would benefit from additional detail regarding AMR testing, using agar dilution method, although authors refer to previous work, they should include a summary of the method. Additionally, critical information regarding availability of raw sequences and the methods used for variant calling, or phylogenetic tree construction are not included.

There are many issues with clarity and formatting issues.

Figure S1 missing a figure legend.

Review Spectrum June 2025

This study by Mehta et al. is a well-designed, technically sound, and epidemiologically insightful analysis of AMR in Kimusu, Kenya. Overall, the study provides significant insights into genomic evolution and the AMR emergence of *N. gonorrhoeae* in the sub-Saharan African region using novel technology. However, the study is limited by the sample collection in a restricted geographical location and the lack of direct biological and behavioral data to support some hypotheses. These gaps are essential to address in follow-up studies. Nonetheless, the study provides a valuable longitudinal genomic analysis of *N. gonorrhoeae* in Kenya, uncovering emerging resistance, linking host factors to AMR evolution, and promoting genomic tools like LIN code for epidemiological surveillance. Below are several strengths and limitations for future considerations:

- The high-resolution approach allows identification of genetic determinants of AMR and lineage tracking.
- The integration of cgMLST and LIN codes improves the resolution and comparability of isolates beyond traditional methods.
- Two-decade comparison (2002–2009 vs 2020–2022) offers a rare longitudinal insight into AMR evolution.
- Comparison with other Kenyan sites and sub-Saharan isolates adds a valuable regional comparison.
- Analysis of HIV status, circumcision status, and age links patients' data to microbial evolution, a novel and valuable epidemiologic dimension.
- Documentation of azithromycin resistance and cephalosporin alert-level MICs is a significant development.
- The authors mention sexual network composition as a possible determinant for the evolution of *N. gonorrhoeae* strains, but social-behavioral data (e.g., partner characteristics, MSM status, condom use) were not collected or analyzed, limiting causal analysis.
- The study proposes that penile microbiome shifts due to VMMC or host inflammation may influence AMR evolution, but this remains highly speculative, as there is no microbiome data, no immunological data, and no causal modeling.

Response to Reviewers

Reviewer 1

This study by Mehta et al. is a well-designed, technically sound, and epidemiologically insightful analysis of AMR in Kimusu, Kenya. Overall, the study provides significant insights into genomic evolution and the AMR emergence of *N. gonorrhoeae* in the sub-Saharan African region using novel technology. However, the study is limited by the sample collection in a restricted geographical location and the lack of direct biological and behavioral data to support some hypotheses. These gaps are essential to address in follow-up studies. Nonetheless, the study provides a valuable longitudinal genomic analysis of *N. gonorrhoeae* in Kenya, uncovering emerging resistance, linking host factors to AMR evolution, and promoting genomic tools like LIN code for epidemiological surveillance. Below are several strengths and limitations for future considerations:

- The high-resolution approach allows identification of genetic determinants of AMR and lineage tracking.
- The integration of cgMLST and LIN codes improves the resolution and comparability of isolates beyond traditional methods.
- Two-decade comparison (2002–2009 vs 2020–2022) offers a rare longitudinal insight into AMR evolution.
- Comparison with other Kenyan sites and sub-Saharan isolates adds a valuable regional comparison.
- Analysis of HIV status, circumcision status, and age links patients' data to microbial evolution, a novel and valuable epidemiologic dimension.
- Documentation of azithromycin resistance and cephalosporin alert-level MICs is a significant development.

→ Thank you for your thoughtful review and constructive comments! Specific responses for areas needing strengthening, explanation, and revision are below.

•The authors mention sexual network composition as a possible determinant for the evolution of *N. gonorrhoeae* strains, but social-behavioral data (e.g., partner characteristics, MSM status, condom use) were not collected or analyzed, limiting causal analysis.

→ We have modified the language to make clearer the explanation and that these are our hypotheses. (lines 421-423, line 438, lines 442-445)

•The study proposes that penile microbiome shifts due to VMMC or host inflammation may influence AMR evolution, but this remains highly speculative, as there is no microbiome data, no immunological data, and no causal modeling.

→ We agree that the hypothesized association is speculative, and have improved the framing of this section to explain why we hypothesize as such. (lines 442-445)

→ Given the associations largely with only circumcision status, we make these hypotheses based on the main features differentiating circumcised and uncircumcised men: (a) Penile microbiome composition [already cited in paper], (b) socioeconomic features. In Kenya, men who have remained uncircumcised tend to be older, with less educational attainment and more economic barriers [1], and therefore likely to have different sexual networks than circumcised men stemming from homophily [2].

→ Therefore, we hypothesize that the underlying features of different gonococcal strains for circumcised and uncircumcised men may relate to penile microbiome composition and/or sexual networks and provide exposition on this. We hope to be able to study this in the future.

Additional citations:

1. Agot K, Onyango J, Otieno G, Musingila P, Gachau S, Ochillo M, Grund J, Joseph R, Mboya E, Ohaga S, Omondi D, Odoyo-June E. Shifting reasons for older men remaining uncircumcised: findings from a pre-and post-demand creation intervention among men aged 25-39 years in western Kenya. *PLoS Glob Public Health* 2024;4(5):e0003188.
2. Kenyon C, Colebunders R. Birds of a feather: homophily and sexual network structure in sub-Saharan Africa. *Int J STD AIDS* 2013;24(3):211-5.

Reviewer #2

Manuscript by Mehta et al., describes genetic determinants of *N. gonorrhoeae* (NG) antimicrobial resistance in Kenya over two decades. The authors examined the changes in circulating NG strains in central regions of Kenya and assessed the possible links between levels of AMR levels, NG type and HIV infection in 193 NG isolates collected between 2002-2009 and then 2020-2022.

While the topic is important and the study adds regional data on NG AMR trends, limitations in data analysis makes it hard to interpret the findings. **The authors do not establish specific links between observed AMR phenotypes and the emergence of known resistance-conferring mutations in the tested isolates.**

- In Table 3, we explicitly present phenotypes (i.e., agar dilution based MICs) and report that we observed the A2059 mutation in all four 23S rRNA alleles of the two isolates with azithromycin resistance and that these two isolates had high predicted MICs. Moreover, there were no A2059 mutations or high predicted MICs for azithromycin isolates, reflecting the specificity of the finding, which we now highlight (line 225).
- We observed elevated agar dilution based MICs for cephalosporins, but did not observe resistance (Table 3), and have clarified this in the Discussion (line 400). Although we found no elevated predicted MICs for cephalosporins, we describe how the *penA* mutations observed (A516G and A501V) are associated with elevated MIC for cephalosporins. The genetic determinants of cefixime and ceftriaxone resistance in NG are not yet completely understood.

Additionally, **the authors did not provide broader insight into other genetic adaptation across the timepoints.**

- The reviewer raises a great question. This would provide novel information, but is outside the scope of the current study. If future funding is obtained, such data mining will be pursued.

Only a relatively small group of related isolates is analyzed in depth. For example, in Fig.1, there are many isolates between two cohorts that share similar ancestral isolate, but they are not discussed, and the genetic/phenotypic changes are not discussed.

- Many isolates do share ancestry, but there was too much sparsity within single STs and lineages to allow for detailed comparison. We therefore abstained from further data drilling due to concern over false signals. As additional data becomes available, this will be something to pursue.

In general, **the main findings are not clear and** the manuscript provides descriptive survey of NG strain types and AMR profiles than a **thorough genomic or evolutionary analysis. A deeper and more robust analysis of genomic data**, other than MLST, phylogeny, and mutations linked to AMR could provide valuable information on pathogenesis, AMR and local epidemiological trends.

- The reviewer's point is well taken in that we did not do formal evolutionary analyses. We have modified the title of the manuscript, and text throughout, to clarify the focus on genetic relatedness. (Title, Abstract lines 2 and 14, Text lines 64 and 346).
- While NG pathogenesis is of interest, it is outside of scope for the current project. As noted in our objectives (lines 65-67), our focus was on AMR determinants. We now reiterate this in the Discussion (lines 347-348).
- Sub-Saharan Africa remains grossly underrepresented in gonococcal surveillance, NG-AMR surveillance, and molecular surveillance, which we note in the limitations. Therefore, providing a descriptive survey as we did is still a large advance in characterizing the epidemiology of NG and NG-AMR in sub-Saharan Africa. We have strengthened this framing in the Background (line 48) and Conclusions.
- We have edited the discussion to clarify the main findings of the study.

The method section would benefit from additional detail regarding AMR testing, using agar dilution method, although authors refer to previous work, they should include a summary of the method.

- Thank you. This has been added. (lines 77-81 and 87-90)

Additionally, critical information regarding availability of raw sequences and the methods used for variant calling, or phylogenetic tree construction are not included.

- As noted in the Data sharing statement, meta-data and whole genome sequencing data are available in PubMLST (isolate IDs are specified) and will be publicly available after manuscript acceptance.
- Details on WGS have been expanded (lines 107-109).
- Details on variant calling and phylogenetic tree construction have been added (lines 115-120).

There are many issues with clarity and formatting issues.

- The authors apologize for not submitting the high resolution images with the initial submission. This has been addressed, and all of the labels have been made larger and file sizes increased.

Figure S1 missing a figure legend.

- This has been added.

Re: Spectrum01586-25R1 (**Genomic Trends and Emerging Antimicrobial Resistance in *Neisseria gonorrhoeae* Over Two Decades in Kenya**)

Dear Dr. Supriya D Mehta:

Thank you for the changes applied to the manuscript following the previous revisions. These changes have significantly improved the quality and while securing reviewers this time has been challenging, we are pleased to accept the manuscript for publication. Thank you for your patience.

Your manuscript has been accepted, and I am forwarding it to the ASM production staff for publication. Your paper will first be checked to make sure all elements meet the technical requirements. ASM staff will contact you if anything needs to be revised before copyediting and production can begin. Otherwise, you will be notified when your proofs are ready to be viewed.

Sincerely,
Monica Cartelle Gestal
Editor
Microbiology Spectrum